# Are Bangladeshi healthcare facilities prepared to provide antenatal care services? Evidence from two nationally representative surveys

Shariful Hakim[1,2], Muhammad Abdul Baker Chowdhury[3], Zobayer Ahmed[4,5], Md Jamal Uddin[1,6]*

1 Department of Statistics, Shahjalal University of Science & Technology, Sylhet, Bangladesh, 2 Chander Hat Degree College, Nilphamari, Bangladesh, 3 Department of Neurosurgery, College of Medicine, University of Florida, Gainesville, Florida, United States of America, 4 Department of Economics, Selcuk University, Selçuklu, Turkey, 5 Department of Economics & Banking, International Islamic University Chittagong, Kumira, Bangladesh, 6 Department of General Educational and Development, Daffodil International University, Dhaka, Bangladesh

* jamal-sta@sust.edu

**Data Availability Statement:** All data is publicly available in dhsprogram website. The link to each survey is available below: https://dhsprogram.com/

## Abstract

Identifying high-risk pregnancies through antenatal care (ANC) is considered the cornerstone to eliminating child deaths and improving maternal health globally. Understanding the factors that influence a healthcare facility's (HCF) preparedness to provide ANC service is essential for assisting maternal and newborn health system progress. We aimed to evaluate the preparedness of HCFs to offer ANC services among childbearing women in Bangladesh and investigate the facility characteristics linked to the preparedness. The data for this study came from two waves of the Bangladesh Health Facilities Survey (BHFS), conducted in 2014 and 2017 using a stratified random sample of facilities. The study samples were 1,508 and 1,506 HCFs from the 2014 and 2017 BHFS, respectively. The outcome variable "ANC services preparedness" was calculated as an index score using a group of tracer indicators. Multinomial logistic regression models were used to identify the significant correlates of ANC service preparedness. We found that private hospitals had a lower chance of having high preparedness than district and upazila public facilities in 2014 (RRR = 0.04, 95% CI: 0.01–0.22, p-value = <0.001) and 2017 (RRR = 0.23, 95% CI: 0.07–0.74, p-value = 0.01), respectively. HCFs from the Khulna division had a 2.84 (RRR = 2.84, CI: 1.25–6.43, p-value = 0.01) and 3.51 (RRR = 3.51, CI: 1.49–8.27, p-value = <0.001) higher likelihood of having medium and high preparedness, respectively, for ANC service compared to the facilities in the Dhaka division in 2017. The facilities that had a medium infection prevention score were 3.10 times (RRR = 3.10, 95% CI: 1.65–5.82; p-value = <0.001) and 1.89 times (RRR = 1.89, 95% CI: 1.09–3.26, p-value = 0.02) more likely to have high preparedness compared to those facilities that had a low infection prevention score in 2014 and 2017 respectively. Facilities without visual aids for client education on pregnancy and ANC were less likely to have high (RRR = 0.29, 95% CI: 0.16–0.53, p-value = <0.001) and (RRR = 0.55, 95% CI: 0.30–0.99, p-value = 0.04) preparedness, respectively, than those with visual aids for client education on pregnancy and ANC in both the surveys. At all two survey time points, facilities that did not maintain individual client cards or records for ANC clients were less likely to

methodology/survey/survey-display-531.cfm
https://dhsprogram.com/methodology/survey/
survey-display-441.cfm.

**Funding:** The authors received no specific funding for this work.

**Competing interests:** The authors have declared that no competing interests exist.

have high (RRR = 0.53, 95% CI: 0.31-.92, p-value = 0.02) and (RRR = 0.41, 95% CI: 0.25–0.66, p-value = <0.001) preparedness, respectively, compared to their counterparts. We conclude that most facilities lack adequate indicators for ANC service preparedness. To improve the readiness of ANC services, government authorities could focus on union-level facilities, community clinics, private facilities, and administrative divisions. They could also make sure that infection control items are available, maintain individual client cards or records for ANC clients, and also ensure ANC clients have access to visual aids.

## Introduction

Pregnancy care, also known as antenatal care (ANC), is an umbrella term utilized to explain the medical care and methods performed on and for childbearing women [1]. It is the care provided to childbearing women by experienced health professionals from the beginning of their pregnancy until the birth of their child. ANC aims to treat the existing problems and those that may arise during pregnancy, affecting the mother and her newborns [2]. Identifying high-risk pregnancies through ANC is considered the cornerstone to eliminating child deaths and improving maternal health globally [3]. The care comprises different tests, preventative treatments, and diagnostic procedures, some of which are performed routinely, and others are offered to the women depending on identified difficulties and risk factors [4]. The quality of health care received during ANC has a considerable impact on the mother's health and pregnancy outcome [2]. According to a World Health Organization (WHO) recommendation, every pregnant woman should receive a minimum of four ANC visits during her pregnancy [5]. They can get the mentioned services by visiting a healthcare facility (HCF) or having health workers come to their homes [6].

In Bangladesh, the number of women who had at least one ANC visit from a qualified health care professional increased from 64% in 2014 to 82% in 2017. During the same period, the percentage of women who had received ≥ 4 ANC visits also rose from 31% to 47%. It demonstrates that Bangladesh is on track to meet the Health, Population, and Nutrition Sector Program's goal of 50% of women completing at least four ANC visits during pregnancy by 2022 [7].

Over the previous few decades, Bangladesh has made significant progress in eliminating maternal mortality, though the fifth target of the Millennium Development Goals (MDG) to cut maternal mortality to 143 deaths per 100,000 live births by 2015 was not fulfilled [8]. Between 1990 and 2015, the maternal mortality ratio (MMR) reduced by 69 percent, from 569 deaths to 176 deaths per 100,000 live births [9]. Bangladesh is one of the nine countries that have made remarkable progress in minimizing maternal mortality by 2015 [10].

In spite of an increase in ANC coverage and a decline in MMR, pregnancy and childbirth-related complications cause about 7,660 annual deaths of reproductive-aged women in Bangladesh [11]. Moreover, maternal mortality accounts for 14% of all reproductive-aged women's deaths [12]. The third sustainable development goal (SDG3) established a global objective, implying that Bangladesh must reduce MMR to less than 70 per 100,000 live births [8]. It is estimated that appropriate medical care at various stages of childbearing difficulty can prevent 90% of maternal deaths and pregnancy-associated diseases [13], and good ANC alone can lessen 20% of maternal mortality [14].

An effective ANC program requires qualified health care providers in a functional health center with referral facilities as well as adequate supplies and diagnostic capabilities [4].

Therefore, a HCF should be well prepared with regard to trained health professionals, medications, supplies, equipment, amenities, and infrastructure to meet this demand [15].

Several studies in Bangladesh have identified the factors responsible for the frequency of ANC visits [14, 16–19]. For instance, the mother's level of education, place of residence, administrative division, media access, and birth order were the most common significant predictors of the frequency of ANC visits in the published studies. The earlier studies that sought to assess ANC service utilization in Bangladesh were limited to rural areas [6, 20, 21]. Some studies identified the level and trends in the frequency of ANC visits [22–24]. But, no research has been conducted in Bangladesh to determine whether or not HCFs are ready to deliver ANC services.

Health facility preparedness is a vital aspect that indicates a facility's commitment to confirming the cumulative availability of items needed to offer a specific service [25]. The evaluation of the preparedness of HCFs for ANC services is essential for both health planning and decision-making to ensure that the mother and baby are healthy at delivery, therefore reducing maternal mortality through improved maternal health. A sound understanding of the factors that influence an HCF's preparedness to provide ANC service is essential for assisting maternal and newborn health system progress. Therefore, we sought to evaluate the preparedness of HCFs to provide ANC services in Bangladesh. We also explored facility characteristics associated with preparedness using two nationally representative surveys.

## Methods

### Study population and setting

This study used publicly available cross-sectional data from two waves of the Bangladesh Health Facilities Survey (BHFS), conducted in 2014 and 2017 using standardized questionnaires from the service provision assessment (SPA) component of the U.S. Agency for International Development (USAID)'s Demographic and Health Surveys (DHS) Program. The National Institute of Population Research and Training (NIPORT) and the Ministry of Health and Family Welfare (MOHFW) implemented the survey with financial assistance from the Government of Bangladesh and USAID with technical help from ICF International, and the Associates for Community and Population Research (ACPR) collected the data. The details of the survey instruments are published elsewhere [26, 27].

The sample for the 2014 and 2017 BHFS was designed to provide results that were representative of all Bangladeshi administrative divisions, and six types of public health facilities were included: district hospitals (DHs), upazila health complexes (UHCs), mother and child welfare centers (MCWCs), union health and family welfare centers (UHFWCs), union sub-centers/rural dispensaries (USC/RDs), and community clinics (CCs). For NGO clinics and hospitals and private hospitals, results are also mentioned separately. Under the Ministry of Health and Family Welfare (MoHFW) in Bangladesh, the highest implementing authority is the Directorate General of Health Services (DGHS). The healthcare infrastructure under the DGHS follows the administrative pattern of Bangladesh, beginning from the national to the district, sub-district, union, and last to the ward levels. It offers encouraging, preventive, and restorative services at various levels, such as primary, secondary, and tertiary [28]. The chart (**Fig 1**) summarizes the hierarchy of Bangladeshi HCFs under DGHS. Within each administrative division, the HCFs were divided by type of facility in both surveys for stratification. The 2014 and 2017 BHFS samples included all kinds of registered health facilities from all administrative divisions across the country. For the 2014 and 2017 BHFS, the sampling frames were a list of 19,184 and 19,811 registered health facilities provided by NIPORT and MOHFW, respectively. A stratified random selection was used to choose 1,596 formal-sector health facilities for the

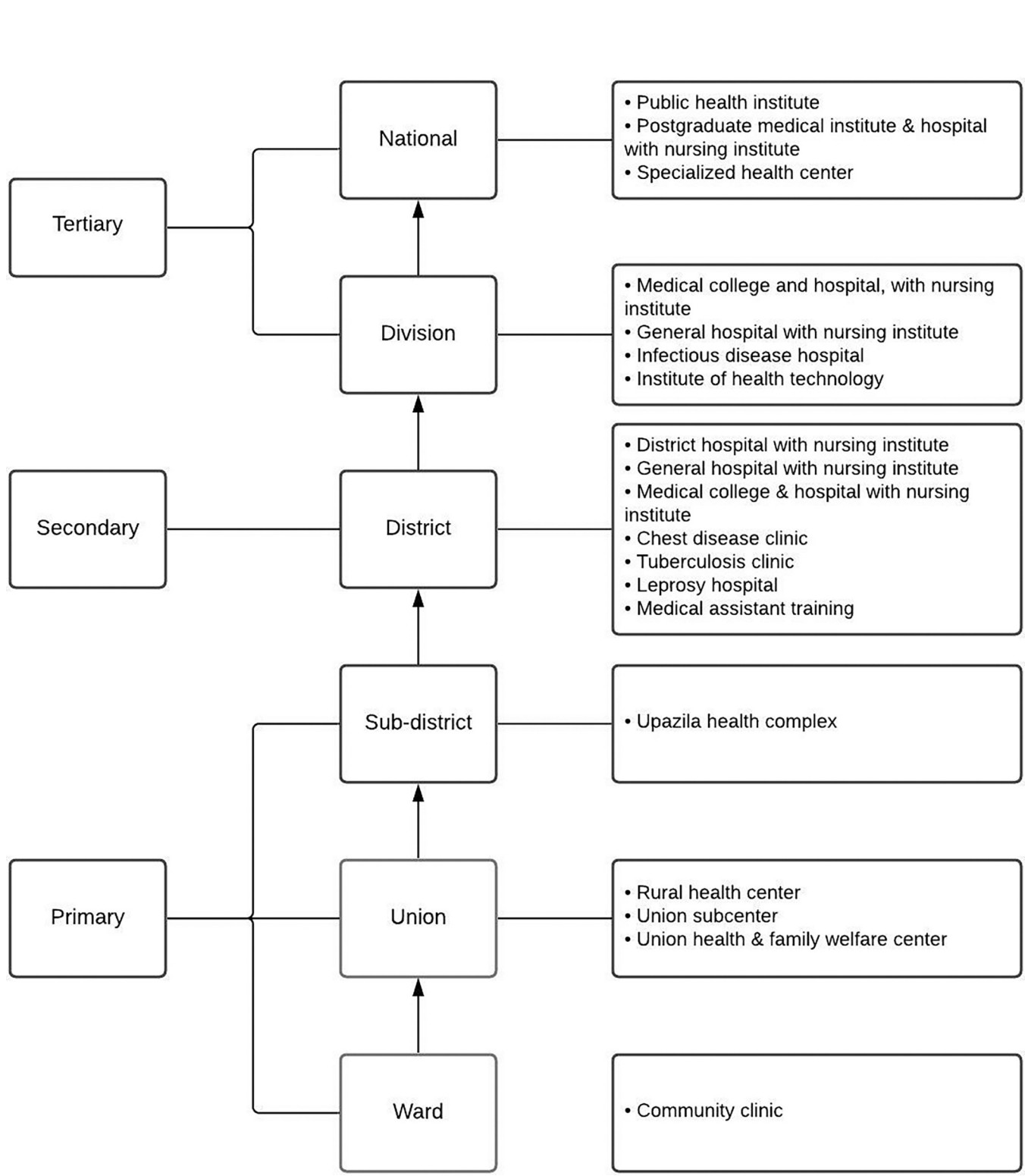

**Fig 1. Hierarchy of Bangladeshi healthcare facilities under the directorate general of health services.**

2014 BHFS and 1,600 for the 2017 BHFS. Some facilities were not surveyed due to different factors, including the fact that they were closed or not operational at the time of the study. Finally, data for 1,548 and 1,524 health facilities are available in the 2014 and 2017 BHFS, respectively. The sampling technique and study design have been described in detail elsewhere [26, 27].

### Study samples

Facilities that provided ANC services were included in the analysis, resulting in 1508 facilities in 2014 BHFS and 1506 facilities in 2017 BHFS.

### Outcome variable

The outcome variable "ANC services preparedness" is defined as the readiness or willingness of the facility to provide ANC services. The WHO has identified a set of tracer indicators that must be available for a health facility to be considered prepared to provide quality ANC services [25]. Availability was defined by the presence of a tracer indicator in a facility that was visible during data collection in accordance with direct observation by the data collectors. We developed a less constrained and Bangladesh-specific-proper version of the WHO-suggested service readiness criteria for ANC service. To meet these criteria, we used six tracer items from the four domains in **Table 1**. The outcome variable was calculated as an index score using the WHO definition of service preparedness index, which considers six indicators for each facility and adds them together to create a mean preparedness score. To obtain a percentage value of the preparedness score, the mean score of the four domains was multiplied by 100.

**Table 1. Measurement procedure of outcome variable "ANC service preparedness".**

| Serial No. | ANC service domain | Tracer indicator | Variable name | Definition | Coding for analysis | ANC service domain score = mean availability of indicators |
|---|---|---|---|---|---|---|
| A | Staff & guidelines | Guideline on ANC | guideline | Availability of national or other ANC related guidelines at the facility | 1 = Available / 0 = Not available | (guideline + trained _staff /2) |
| | | Staff trained in ANC | trained _staff | Availability of at least one staff member offering ANC service ever trained in some point of ANC | 1 = Available / 0 = Not available | |
| B | Equipment | Blood pressure apparatus | bp | Availability of digital blood pressure machine or manual sphygmomanometer with stethoscope | 1 = Available / 0 = Not available | bp |
| C | Diagnostic capability | Hemoglobin test | hemoglobin | Availability of hemoglobin test for clients at the facility | 1 = Available / 0 = Not available | (hemoglobin + urine_protein/2) |
| | | Urine protein test | urine_protein | Availability of urine protein test for clients at the facility | 1 = Available / 0 = Not available | |
| D | Medicines | Iron or folic acid tablets | iron _folic | Availability of at least one valid dose of any of the following 3: Iron tablet Folic acid tablet Combined iron and folic acid tablets | 1 = Available / 0 = Not available | iron _folic |
| ANC service preparedness index | | | | | | Mean score of the four domains = (A+B+C+D)/4 |

**Source:** Service availability and readiness assessment (SARA) implementation guide [30]

Mathematically, the preparedness score for the *pth* facility can be expressed as follows [29]:

$$D_p = \left( \frac{\sum_{q=1}^{t} \frac{1}{w_q} \sum_{r=1}^{w_q} y_{pqr}}{t} \right) \times 100$$

$$p = 1, 2, \ldots, s; q = 1, 2, \ldots, t; r = 1, 2, \ldots, w_q$$

where, $D_p$ = the score of the pth HCF

$t$ = the number of domains

$w_q$ = the number of indicators in the qth domain

$y_{pqr}$ = is the value of rth indicator in qth domain for pth HCF

$s$ = total number of HCF

Hence, this preparedness index is a continuous variable whose values lie between 0 and 100. The higher the readiness score, the more prepared the facility is for ANC services, whereas the lower the score, the less prepared it is [30]. For example, if a facility has a score of 100, it means that the facility is fully ready to offer ANC service. If a facility has a score of 50, it is only half prepared to provide the service. The distribution of preparedness scores was divided into three equal sections using the 33rd and 66th percentiles. Low, medium and high preparedness for providing ANC services were defined as facilities with a preparedness score of less than 62.5percent (33rd percentile), between 62.5 percent (33rd percentile) and 75.0 percent (66th percentile), and more than 75.0 percent (66th percentile). In previous investigations [31–33], the authors used this categorization concept. Mathematically, the outcome variable of this study can be expressed as follows [29]:

$$U_p = \begin{cases} 0, & \text{Low if } D_p < P_{33} = 62.5 \\ 1, & \text{Medium if } P_{33} = 62.5 \leq D_p < P_{66} = 75.0 \\ 2, & \text{High, if } D_p \geq P_{66} = 75.0 \end{cases}$$

where, $U_p$ = the ordinal response of the pth HCF

$P_{33}$ = 33rd percentile of $D_p$

$P_{66}$ = 66th percentile of $D_p$

## Potential factors

Based on prior studies on the preparedness of HCF, we chose 17 possible factors to find out how these factors correlated with the preparedness of facilities to offer ANC services [34–36]. Potential factors of interest include facility type (district and upazila public facilities, union-level public facilities, public community clinic (CC), non-government organization (NGO) clinic/hospital, private hospital), managing authority i.e. ownership of facility (public, private), location (urban, rural), administrative division (Barisal, Chittagong, Dhaka, Khulna, Rajshahi,

Rangpur, Sylhet, Mymensingh), quality assurance activities (performed, not performed), external supervision (received, not received), routine user fee (not available, available), 24-hour staff coverage (not available, available), inpatient capability (yes, no), number of days per month ANC services provided (provides but not every day, provides every day), visual aids for client education related to pregnancy/ANC (available, not available), sufficient privacy for ANC exam (available, not available), infection safety precaution guideline (available, not available), individual client cards or records for ANC clients (maintained, not maintained), basic amenities score (low, medium, high), infection prevention score (low, medium, high).

Regular electricity, improved water source, visual and auditory privacy during consultations, client latrine, communication equipment (landline/mobile phone), and computer with internet access were among the basic amenities included in the domain score. Moreover, the infection prevention score contained the following tracer items: waste bin, sharps box, general disinfectant, syringes/needles, sterile disposable gloves, hand hygiene, running water, soap/hand disinfectant. The score for every facility is calculated by dividing the total number of tracer items by the sum of their availabilities (i.e., value = 1). Each scoring variable is a continuous number that was categorized into three equal segments using 33rd and 66th percentiles. Low, medium, and high basic amenities scores were defined as facilities with an amenity score of less than 33.3 percent (33rd percentile), between 33.3 (33rd percentile) and 66.7 percent (66th percentile), and more than 66.7 percent (66th percentile). Furthermore, facilities with infection scores of less than 50% (33rd percentile), 50% (33rd percentile), 87.5 percent (66th percentile), and more than 87.5 percent (66th percentile) had low, medium, and high infection prevention scores.

## Data analysis

Using the chi-square test, we compared the proportion of ANC service preparedness between the categories of various potential factors. Since the outcome variable "ANC service preparedness" is a polytomous variable, a multinomial logistic regression model was used to identify the characteristics of the HCF that are linked to ANC service readiness. It took low preparedness as the baseline outcome category. We omitted the variable "external supervisory visit to a facility" from the final model due to its lower frequency in one category. Moreover, in the 2014 and 2017 surveys, there were 15 and 14 missing values in the "Routine quality assurance activities" variable and these values were left out of the regression modeling. Multicollinearity was assessed using the variance inflation factor (VIF), and we removed two variables-managing authority and inpatient care- from the final model due to VIF > 3. We performed all data management and analyses using Stata 13 (StataCorp, College Station, TX, USA). To accommodate the complicated survey design, we used the weight option in STATA with the sampling weights given in the dataset to weight all of our analyses. When we did the modeling exercise, we used the "svy" command in STATA to consider the survey design, primary sampling unit, and cluster.

## Ethics statement

This study was based on an analysis of the 2017 and 2014 BHFS survey datasets that are already in the public domain and can be found online freely. The Institutional Review Boards (IRB) of ICF International and the Bangladesh Medical Research Council of the Ministry of Health and Family Welfare (MOH & FW) have evaluated and approved the methodology and questionnaires for the surveys. The informed consent was solicited and gained from the manager, the man in charge of the facility, or the facility's most high-ranking health worker accountable for client services. All the important features of the study, including its purpose and interview

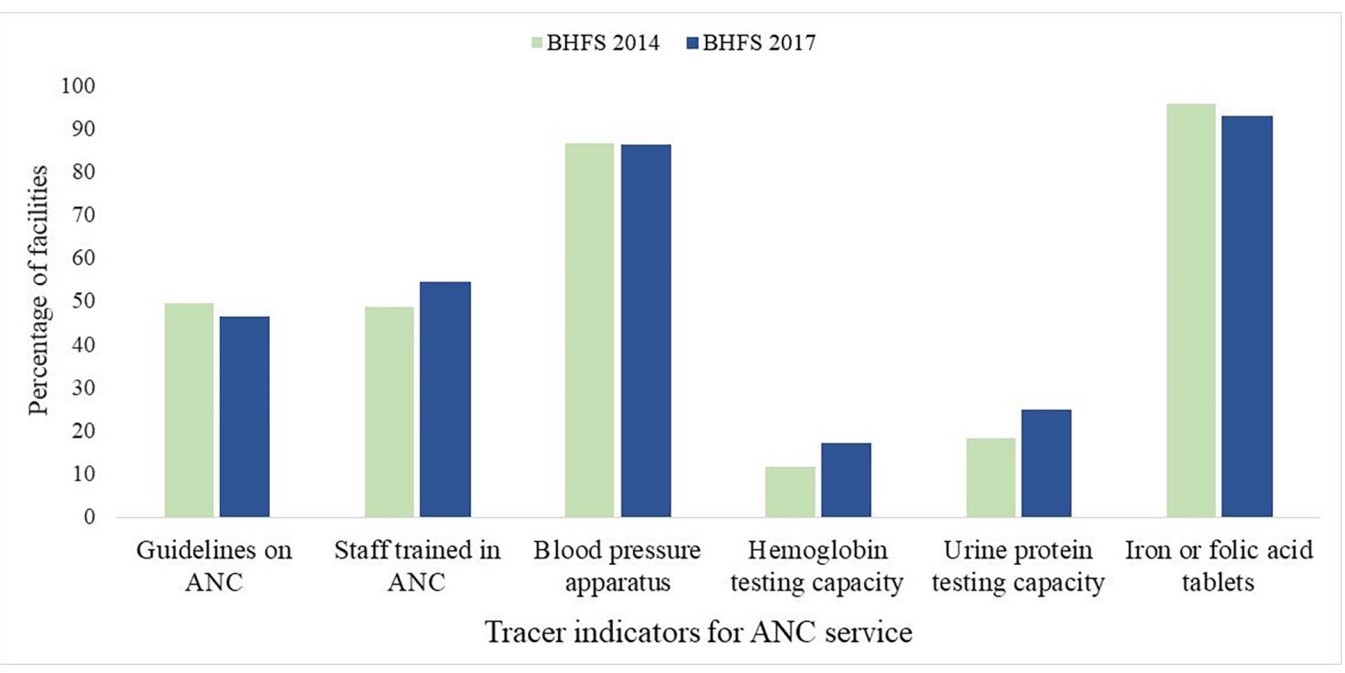

**Fig 2. Availability of tracer indicators in healthcare facilities measuring preparedness to offer ANC service by survey year.**

procedures, were adequately explained to the respondents. Respondents whose facilities agreed to take part in the study provided a written consent where they signed. In addition, the DHS has permitted us to use the datasets. So, this study didn't need to get any more ethical approval.

## Results

### Univariate analysis

The percentage of facilities offering ANC services has risen slightly from 97.4 percent in 2014 to 98.8 percent in 2017. Of the facilities that provide ANC services, only 4.4% and 4.3% have all tracer indicators (i.e., they are completely prepared) to provide quality ANC in 2014 and 2017, respectively (**S1 Fig**). **Fig 2** shows the availability of tracer indicators for ANC service preparedness (i.e., among facilities that provide ANC services, the percentages of the tracer items that were observed to be available on the day of the assessment at the service site). Since 2014, the percentage of facilities providing ANC services with at least one staff member trained in ANC has increased (from 48.7% to 54.6%), as could be seen from the conducted hemoglobin tests (from 11.7% to 17.2%) and urine protein tests (from 18.5% to 24.8%). However, there has been a decline in the percentage of facilities providing ANC service with the availability of ANC guidelines (from 49.6% to 46.4%), blood pressure equipment (from 86.7% to 86.4%), and iron or folic acid tablets (from 95.7% t to 92.9%). **Table 2** displays the availability of tracer indicators for ANC service preparation by facility type. Between 2014 and 2017, the availability of ANC guidelines in private hospitals decreased from 24.7% to 6.6%. In every facility type, the capacity to carry out hemoglobin and urine protein testing has improved, whereas the availability of iron or folic acid tablets has declined over the past three years.

### Bivariate analysis

**Table 3** depicts the proportion of ANC service preparedness between the categories of various potential factors. In 2014, 36.4% of union-level facilities and 36.9% of rural facilities had low

**Table 2. Distribution of tracer indicators for ANC service preparedness by facility type.**

| Facility type | Guideline on ANC, n (%) | | Staff trained for ANC, n (%) | | Blood pressure apparatus, n (%) | | Hemoglobin testing capacity, n (%) | | Urine protein testing capacity, n (%) | | Iron or folic acid tablets, n (%) | |
|---|---|---|---|---|---|---|---|---|---|---|---|---|
| | BHFS 2014 | BHFS 2017 | BHFS 2014 | BHFS 2017 | BHFS 2014 | BHFS 2017 | BHFS 2014 | BHFS 2017 | BHFS 2014 | BHFS 2017 | BHFS 2014 | BHFS 2017 |
| District and upazila public facilities | 33 (70.3) | 26 (60.3) | 39 (83.6) | 37 (85.5) | 46 (97.7) | 43 (98.8) | 33 (70.2) | 33 (74.4) | 32 (68.1) | 29 (66.1) | 47 (99.1) | 42 (96.1) |
| Union-level public facilities | 170 (47.8) | 174 (50.4) | 178 (50.0) | 185 (53.5) | 304 (85.4) | 313 (90.6) | 28 (7.9) | 52 (15.0) | 43 (12.0) | 63 (18.1) | 346 (97.3) | 327 (94.5) |
| Public community clinic | 480 (48.2) | 447 (44.2) | 462 (46.3) | 549 (54.2) | 853 (85.5) | 842 (83.2) | 36 (3.6) | 93 (9.2) | 123 (12.3) | 196(19.4) | 952 (95.4) | 940 (92.8) |
| NGO clinic/hospital | 58 (72.9) | 48 (75.1) | 48 (60.4) | 40 (63.8) | 78 (98.3) | 62 (98.0) | 55 (69.3) | 47 (74.8) | 60 (75.1) | 53 (83.7) | 75 (94.8) | 58 (91.5) |
| Private hospital | 7 (24.7) | 3 (6.6) | 8 (27.5) | 11 (26.4) | 26 (94.7) | 40 (97.7) | 24 (87.3) | 33 (81.6) | 22 (78.5) | 32 (79.3) | 23 (81.7) | 32 (78.0) |

preparedness to offer ANC services, compared to 30.3% and 33.0% in 2017. Between 2014 and 2017, the medium level of ANC service preparedness increased among facilities that received an external supervisory visit six months before the survey (from 26.8% to 29.0%) and had 24-hour staff coverage (from 15.1% to 19.5%). Moreover, there have been only minimal decreases in the high preparedness of ANC service since 2014 among facilities that performed quality assurance activities (from 55.2% to 39.7%) and had separate user fees or charges for each client services (from 81.8% to 63.8%).

## Multivariable analysis

The associations between the potential factors and ANC service preparedness, obtained from the multinomial logistic regression models fitted to the medium and high categories compared with low, are presented in **Table 4**. The significant factors of medium and low preparedness changed throughout the survey years. In 2014, the chance of having medium preparedness was 80% less for private facilities as compared to the district and public facilities (Relative risk ratio (RRR) = 0.08, 95% Confidence interval (CI): 0.01–0.47; p-value = 0.01). Similarly, private hospitals had lower odds of having high preparedness than district and upazila public facilities in 2014 (RRR = 0.04, 95% CI: 0.01–0.22, p-value = <0.001) and 2017 (RRR = 0.23, 95% CI: 0.07–0.74, p-value = 0.01), respectively. Moreover, in 2017, union-level public facilities and CCs had an 86% (RRR = 0.14, 95% CI: 0.06–0.36, p-value = <0.001) and 88% (RRR = 0.12, 95% CI: 0.04–0.34, p-value = <0.001) lower probability of being highly prepared to provide ANC service compared to district and upazila level public facilities. In terms of administrative division, it was not significantly associated with ANC service preparedness in 2014. But HCFs from the Khulna division had a 2.84 (RRR = 2.84, CI: 1.25–6.43, p-value = 0.01) and 3.51 (RRR = 3.51, CI: 1.49–8.27, p-value = <0.001) higher likelihood of having medium and high preparedness respectively for ANC service compared to the facilities in the Dhaka division in 2017. The facilities that had a medium infection prevention score were 3.10 times (RRR = 3.10, 95% CI: 1.65–5.82; p-value = <0.001) and 1.89 times (RRR = 1.89, 95% CI: 1.09–3.26, p-value = 0.02) more likely to have high preparedness compared to those facilities that had a low infection prevention score in 2014 and 2017 respectively. Facilities without visual aids for client education on pregnancy and ANC were less likely to have high (RRR = 0.29, 95% CI: 0.16–0.53, p-value = <0.001) and (RRR = 0.55, 95% CI: 0.30–0.99, p-value = 0.04) preparedness, respectively, than those with visual aids for client education on pregnancy and ANC in both the surveys. At all two survey time points, facilities that did not maintain individual client cards or records for ANC clients were less likely to have high (RRR = 0.53, 95% CI: 0.31-.92, p-value = 0.02) and

**Table 3. Distribution of different factors and their associations with ANC service preparedness of Bangladeshi health care facilities by survey years.**

| Factors | BHFS 2014 (n = 1,508) | | | | BHFS 2017 (n = 1,506) | | | |
| --- | --- | --- | --- | --- | --- | --- | --- | --- |
| | ANC service preparedness | | | P-value* | ANC service preparedness | | | |
| | Low | Medium | High | | Low | Medium | High | P-value* |
| **Facility type** | | | | <0.001 | | | | <0.001 |
| District and upazila public facilities | 1 (2.8) | 5 (9.9) | 41 (87.3) | | 2 (4) | 5 (10.4) | 38 (85.6) | |
| Union-level public facilities | 130 (36.4) | 121 (33.8) | 106 (29.7) | | 105 (30.3) | 114 (32.9) | 127 (36.8) | |
| Public community clinic | 381 (38.2) | 360 (36.1) | 257 (25.8) | | 356 (35.1) | 358 (35.4) | 299 (29.5) | |
| NGO clinic | 5 (6.8) | 7 (9.2) | 67 (84) | | 2 (3.6) | 6 (8.9) | 55 (87.5) | |
| Private hospital | 6 (21) | 3 (12.4) | 19 (66.6) | | 8 (20.6) | 7 (16.5) | 26 (62.9) | |
| **Managing authority (Ownership of facility)** | | | | <0.001 | | | | <0.001 |
| Public | 512 (36.5) | 485 (34.6) | 404 (28.8) | | 462 (33) | 476 (34) | 464 (33.1) | |
| Private | 11 (10.5) | 11 (10) | 85 (79.5) | | 11 (10.2) | 12 (11.9) | 81 (77.9) | |
| **Location of facility** | | | | <0.001 | | | | <0.001 |
| Urban | 12 (9.7) | 13 (11) | 96 (79.4) | | 10 (9.9) | 13 (11.9) | 82 (78.2) | |
| Rural | 511 (36.9) | 483 (34.8) | 393 (28.4) | | 462 (33) | 476 (34) | 462 (33) | |
| **Administrative division of facility** | | | | <0.001 | | | | <0.001 |
| Barishal | 38 (33.3) | 50 (43.9) | 26 (22.9) | | 43 (37.9) | 29 (26.1) | 40 (36) | |
| Chattogram | 115 (41.1) | 82 (29.4) | 82 (29.5) | | 91 (32.2) | 77 (27) | 116 (40.9) | |
| Dhaka | 136 (33.1) | 138 (33.5) | 138 (33.4) | | 122 (40.5) | 86 (28.8) | 92 (30.7) | |
| Khulna | 58 (30.1) | 80 (41.8) | 54 (28) | | 35 (19.2) | 67 (36.3) | 82 (44.4) | |
| Rajshahi | 98 (46.1) | 56 (26.4) | 58 (27.4) | | 66 (30.7) | 67 (31.3) | 81 (38) | |
| Rangpur | 51 (25.2) | 54 (26.6) | 98 (48.2) | | 57 (29.7) | 92 (48.2) | 42 (22.1) | |
| Sylhet | 27 (28.2) | 35 (36.8) | 33 (35) | | 29 (30.1) | 31 (32) | 36 (37.9) | |
| Mymensingh | | | | | 30 (24.6) | 39 (31.8) | 53 (43.7) | |
| **Routine quality assurance activities** | | | | <0.001 | | | | 0.01 |
| Not performed | 465 (37.8) | 422 (34.3) | 343 (27.9) | | 290 (30.7) | 332 (35.1) | 322 (34.1) | |
| Performed | 53 (20.2) | 64 (24.6) | 145 (55.2) | | 177 (32.4) | 153 (27.9) | 218 (39.7) | |
| **External supervisory visit to facility** | | | | 0.04 | | | | <0.001 |
| Not received | 21 (44.5) | 8 (16.1) | 19 (39.4) | | 13 (76.7) | 0 (1.4) | 4 (21.8) | |
| Received, within the past 6 months | 457 (33.7) | 460 (33.9) | 439 (32.4) | | 419 (30) | 461 (33.1) | 515 (36.9) | |
| Received, more than 6 months ago | 45 (43.1) | 28 (26.8) | 31 (30) | | 41 (43.7) | 27 (29) | 26 (27.3) | |
| **Routine user fee or charges for client service** | | | | <0.001 | | | | <0.001 |
| No routine user fee | 504 (37.2) | 475 (35.1) | 375 (27.7) | | 295 (34.9) | 299 (35.5) | 250 (29.6) | |
| Yes, fixed for all service | 9 (20.6) | 10 (23.9) | 24 (55.4) | | 133 (32.6) | 142 (34.8) | 133 (32.6) | |
| Yes, separate for each service | 10 (8.7) | 10 (9.5) | 90 (81.8) | | 45 (17.7) | 47 (18.5) | 162 (63.8) | |
| **24-hours staff coverage** | | | | <0.001 | | | | <0.001 |
| Not available | 503 (37.8) | 469 (35.3) | 358 (26.9) | | 428 (33.7) | 443 (34.9) | 399 (31.5) | |
| Available | 20 (11.1) | 27 (15.1) | 132 (73.8) | | 45 (19.1) | 46 (19.5) | 145 (61.4) | |
| **Inpatient care** | | | | <0.001 | | | | <0.001 |
| Not available | 510 (36.4) | 481 (34.4) | 409 (29.2) | | 459 (32.9) | 474 (34) | 464 (33.2) | |
| Available | 13 (12.2) | 15 (13.6) | 81 (74.2) | | 14 (12.7) | 14 (13.1) | 81 (74.2) | |
| **Basic amenities score** | | | | <0.001 | | | | <0.001 |
| Low | 169 (38.7) | 177 (40.5) | 90 (20.7) | | 51 (39.1) | 58 (44.9) | 21 (16) | |
| Medium | 326 (39.4) | 259 (31.3) | 241 (29.2) | | 330 (34.8) | 297 (31.3) | 321 (33.9) | |
| High | 28 (11.5) | 60 (24.5) | 158 (64) | | 93 (21.6) | 134 (31.1) | 203 (47.3) | |
| **Infection prevention score** | | | | <0.001 | | | | <0.001 |
| Low | 270 (53.7) | 171 (34.1) | 62 (12.3) | | 166 (42.4) | 141 (35.8) | 86 (21.8) | |
| Medium | 197 (32.3) | 218 (35.8) | 194 (31.8) | | 233 (29.3) | 264 (33.2) | 298 (37.5) | |
| High | 56 (14.1) | 106 (26.8) | 233 (59) | | 73 (23) | 84 (26.4) | 161 (50.6) | |
| **Availability of infection safety precaution guideline** | | | | <0.001 | | | | <0.001 |
| Available | 56 (14.6) | 119 (30.8) | 210 (54.6) | | 114 (26.9) | 118 (27.8) | 192 (45.2) | |
| Not available | 467 (41.6) | 377 (33.6) | 279 (24.8) | | 359 (33.1) | 371 (34.3) | 353 (32.6) | |
| **Number of days per month ANC services provided** | | | | 0.01 | | | | 0.18 |
| Provides but not everyday | 100 (41.7) | 81 (34) | 58 (24.3) | | 33 (24.9) | 45 (33.8) | 56 (41.4) | |
| Provides everyday | 423 (33.3) | 414 (32.7) | 431 (34) | | 439 (32) | 443 (32.3) | 489 (35.7) | |

*(Continued)*

**Table 3.** (Continued)

| Factors | BHFS 2014 (n = 1,508) | | | P-value* | BHFS 2017 (n = 1,506) | | | |
| --- | --- | --- | --- | --- | --- | --- | --- | --- |
| | ANC service preparedness | | | | ANC service preparedness | | | |
| | Low | Medium | High | | Low | Medium | High | P-value* |
| **Visual aid for client education related to pregnancy/ANC** | | | | <0.001 | | | | 0.01 |
| Available | 260 (25.8) | 343 (34) | 406 (40.2) | | 362 (29.5) | 398 (32.4) | 468 (38.1) | |
| Not available | 263 (52.7) | 153 (30.7) | 83 (16.7) | | 111 (39.8) | 91 (32.7) | 76 (27.4) | |
| **Sufficient privacy for ANC exam** | | | | <0.001 | | | | 0.16 |
| Available | 176 (28.4) | 188 (30.4) | 255 (41.2) | | 341 (31.8) | 333 (31) | 399 (37.2) | |
| Not available | 347 (39) | 308 (34.6) | 234 (26.4) | | 131 (30.3) | 156 (36) | 146 (33.7) | |
| **Individual client cards or records for ANC clients** | | | | <0.001 | | | | <0.001 |
| Maintained | 196 (23.9) | 282 (34.6) | 339 (41.5) | | 227 (24.7) | 293 (32) | 397 (43.3) | |
| Not maintained | 327 (47.4) | 213 (30.9) | 150 (21.7) | | 246 (41.8) | 195 (33.2) | 147 (25) | |

Note: Due to the rounding of cell counts, percentages may not add to 100% and number of valid cases may be dissimilar to the total count. Frequencies and percentages indicate weighted data.

* P-value for χ2 test

(RRR = 0.41, 95% CI: 0.25–0.66, p-value = <0.001) preparedness, respectively, compared to their counterparts.

## Discussion

Using two waves (2014 and 2017) of nationally representative survey data from Bangladesh, the purpose of this study is to investigate HCFs' preparedness to provide ANC services to pregnant women and identify health facility features connected to the preparedness. The main findings are as follows: 1) There were no significant changes in overall ANC readiness between 2014 and 2017; 2) About one out of every twenty-five facilities is fully prepared to offer quality ANC services; 3) About half of the HCFs had staff who received in-service ANC training and guidelines on ANC service; 4) Throughout all survey periods, only a small percentage of HCFs that provide ANC services can perform a hemoglobin or urine protein test; 5) Overall, the significant factors of medium and low preparedness: Privately owned facilities and facilities with adequate infection prevention items were significantly associated with both medium and high preparedness in 2014 but only high preparedness in 2017. There was a significant link between administrative division and ANC service preparedness in 2017, with HCF from the Khulna and Rangpur divisions being more likely to have medium or high preparedness to provide ANC services than those from the Dhaka division. The availability of visual aid for client education related to pregnancy/ANC and maintaining individual client cards or records for ANC clients significantly correlated with the medium and low preparedness in the 2014 and 2017 BHFS. Furthermore, there is a significant link between ANC service preparedness and the availability of infection safety precaution guidelines in 2014.

Most facilities lack adequate indicators for ANC service preparedness, with just 3 facilities out of 75 being fully prepared to offer quality ANC service. Findings associated with the preparedness of HCFs to offer quality ANC service are compatible with prior studies in Ethiopia

**Table 4. Factors associated with ANC service preparedness: Relative risk ratio (RRR), 95% confidence intervals (CI), and P-values from adjusted multinomial logistic regression model by the survey years.**

| Factors | BHFS 2014 | | | | BHFS 2017 | | | |
| --- | --- | --- | --- | --- | --- | --- | --- | --- |
| | ANC service preparedness | | | | ANC service preparedness | | | |
| | Medium versus low | | High versus low | | Medium versus low | | High versus low | |
| | RRR (95% CI) | P-value | RRR (95% CI) | P-value | RRR (95% CI) | P-value | RRR (95% CI) | P-value |
| **Facility type** | | | | | | | | |
| District and upazila public facilities | Reference | | Reference | | Reference | | Reference | |
| Union-level public facilities | 1.43 (0.33, 6.12) | 0.63 | 0.66 (0.15, 2.79) | 0.57 | 0.58 (0.19, 1.81) | 0.35 | 0.14 (0.06, 0.36) | <0.001 |
| Public community clinic | 1.63 (0.38, 7.07) | 0.51 | 0.84 (0.18, 3.9) | 0.82 | 0.52 (0.16, 1.7) | 0.28 | 0.12 (0.04, 0.34) | <0.001 |
| NGO clinic | 0.46 (0.07, 3.09) | 0.43 | 0.67 (0.11, 4.14) | 0.67 | 1.06 (0.19, 5.92) | 0.95 | 1.55 (0.28, 8.6) | 0.62 |
| Private hospital | 0.08 (0.01, 0.47) | 0.01 | 0.04 (0.01, 0.22) | <0.001 | 0.37 (0.1, 1.47) | 0.16 | 0.23 (0.07, 0.74) | 0.01 |
| **Location of facility** | | | | | | | | |
| Urban | Reference | | Reference | | Reference | | Reference | |
| Rural | 0.53 (0.13, 2.06) | 0.36 | 0.48 (0.11, 2.07) | 0.33 | 0.69 (0.16, 3.02) | 0.62 | 0.52 (0.17, 1.62) | 0.26 |
| **Administrative division of facility** | | | | | | | | |
| Barishal | 1.10 (0.52, 2.33) | 0.80 | 0.53 (0.23, 1.20) | 0.13 | 1.08 (0.47, 2.48) | 0.85 | 2.12 (0.95, 4.75) | 0.07 |
| Chattogram | 0.69 (0.34, 1.41) | 0.31 | 0.75 (0.35, 1.63) | 0.47 | 1.34 (0.62, 2.88) | 0.45 | 2.74 (1.3, 5.77) | 0.01 |
| Dhaka | Reference | | Reference | | Reference | | Reference | |
| Khulna | 1.48 (0.68, 3.21) | 0.32 | 1.08 (0.41, 2.86) | 0.87 | 2.84 (1.25, 6.43) | 0.01 | 3.51 (1.49, 8.27) | <0.001 |
| Rajshahi | 0.55 (0.25, 1.23) | 0.14 | 0.66 (0.28, 1.58) | 0.35 | 1.65 (0.71, 3.87) | 0.25 | 2.6 (1.09, 6.15) | 0.03 |
| Rangpur | 0.80 (0.32, 1.99) | 0.63 | 1.23 (0.46, 3.28) | 0.67 | 2.77 (1.29, 5.91) | 0.01 | 1.52 (0.65, 3.56) | 0.33 |
| Sylhet | 0.84 (0.36, 1.96) | 0.69 | 0.76 (0.32, 1.84) | 0.54 | 1.46 (0.6, 3.58) | 0.4 | 2.19 (0.88, 5.44) | 0.09 |
| Mymensingh | | | | | 1.93 (0.80, 4.64) | 0.14 | 3.51 (1.47, 8.40) | <0.001 |
| **Routine quality assurance activities** | | | | | | | | |
| Not performed | Reference | | Reference | | Reference | | Reference | |
| Performed | 2.02 (0.75, 5.42) | 0.16 | 7.36 (2.2, 24.65) | <0.001 | 1.04 (0.56, 1.93) | 0.91 | 2.05 (1.08, 3.89) | 0.03 |
| **Routine user fee or charges for client service** | | | | | | | | |
| Not available | Reference | | Reference | | Reference | | Reference | |
| Available | 1.16 (0.30, 4.57) | 0.83 | 2.65 (0.78, 8.97) | 0.12 | 0.97 (0.58, 1.61) | 0.89 | 1.38 (0.82, 2.31) | 0.22 |
| **24-hours staff coverage** | | | | | | | | |
| Not available | Reference | | Reference | | Reference | | Reference | |
| Available | 1.28 (0.62, 2.64) | 0.51 | 1.73 (0.84, 3.57) | 0.14 | 0.71 (0.44, 1.13) | 0.14 | 0.74 (0.45, 1.21) | 0.23 |
| **Basic amenities score** | | | | | | | | |
| Low | Reference | | Reference | | Reference | | Reference | |
| Medium | 0.66 (0.39, 1.13) | 0.13 | 0.83 (0.44, 1.58) | 0.57 | 0.87 (0.39, 1.93) | 0.72 | 2.99 (1.19, 7.48) | 0.02 |

(*Continued*)

**Table 4.** (Continued)

| Factors | BHFS 2014 | | | | BHFS 2017 | | | |
| --- | --- | --- | --- | --- | --- | --- | --- | --- |
| | ANC service preparedness | | | | ANC service preparedness | | | |
| | Medium versus low | | High versus low | | Medium versus low | | High versus low | |
| | RRR (95% CI) | P-value | RRR (95% CI) | P-value | RRR (95% CI) | P-value | RRR (95% CI) | P-value |
| High | 1.62 (0.53, 5) | 0.4 | 1.65 (0.47, 5.86) | 0.44 | 1.71 (0.65, 4.51) | 0.28 | 5.3 (1.81, 15.54) | <0.001 |
| **Infection prevention score** | | | | | | | | |
| Low | Reference | | Reference | | Reference | | Reference | |
| Medium | 1.62 (0.98, 2.69) | 0.06 | 3.1 (1.65, 5.82) | <0.001 | 1.32 (0.8, 2.18) | 0.28 | 1.89 (1.09, 3.26) | 0.02 |
| High | 2.08 (1.01, 4.3) | 0.05 | 6.43 (3.04, 13.59) | <0.001 | 1.19 (0.62, 2.27) | 0.6 | 1.72 (0.86, 3.45) | 0.13 |
| **Infection safety precaution guideline** | | | | | | | | |
| Available | Reference | | Reference | | Reference | | Reference | |
| Not available | 0.54 (0.31, 0.93) | 0.03 | 0.46 (0.25, 0.84) | 0.01 | 1.13 (0.67, 1.89) | 0.65 | 0.81 (0.49, 1.36) | 0.43 |
| **Number of days per month ANC services provided** | | | | | | | | |
| Provides but not everyday | Reference | | Reference | | Reference | | Reference | |
| Provides everyday | 0.99 (0.55, 1.77) | 0.97 | 1.04 (0.54, 2.01) | 0.9 | 0.91 (0.44, 1.87) | 0.80 | 0.59 (0.27, 1.29) | 0.19 |
| **Visual aid for client education related to pregnancy/ANC** | | | | | | | | |
| Available | Reference | | Reference | | Reference | | Reference | |
| Not available | 0.59 (0.36, 0.97) | 0.04 | 0.29 (0.16, 0.53) | <0.001 | 0.82 (0.47, 1.43) | 0.48 | 0.55 (0.3, 0.99) | 0.04 |
| **Sufficient privacy for ANC exam** | | | | | | | | |
| Available | Reference | | Reference | | Reference | | Reference | |
| Not available | 0.77 (0.47, 1.27) | 0.31 | 0.72 (0.40, 1.31) | 0.28 | 1.5 (0.84, 2.65) | 0.17 | 2.34 (1.29, 4.25) | 0.01 |
| **Individual client cards or records for ANC clients** | | | | | | | | |
| Maintained | Reference | | Reference | | Reference | | Reference | |
| Not maintained | 0.56 (0.35, 0.91) | 0.02 | 0.53 (0.31, 0.92) | 0.02 | 0.67 (0.42, 1.06) | 0.09 | 0.41 (0.25, 0.66) | <0.001 |

and Nepal that found the facilities were unable to satisfy the needs set by WHO recommendations [15, 37]. Furthermore, a study conducted in India found that ANC services in the surveyed HCFs were of poor quality, even though this study explained the quality of ANC services in a variety of ways, including the ANC utilization, clinical quality, and interpersonal quality of care measures [38]. The study utilized the availability of infrastructure, essential medicines, necessary equipment, laboratory supplies, and vaccines as quality measures. However, we used a WHO guideline in this study to assess the structural preparedness to evaluate the service availability and preparedness of HCFs. Another multi-country study on quality ANC preparedness in Sub-Saharan Africa revealed the preparedness in particular tracer items of ANC services, where, in terms of ANC guidelines, trained staff, and iron supplementation, the majority of the countries were more prepared than Bangladesh [39]. Routine and robust monitoring of Bangladesh's HCFs is necessary to ensure complete ANC preparedness among the facilities.

Assuring that all facilities have access to ANC service guidelines for health personnel, which will aid in checking that protocols are followed, is one step toward evaluating the quality of

ANC service [34]. But, in this study, such guidelines were lacking in most private facilities. Moreover, this study found that extensive training on any one of the ANC topics (ANC screening, counseling, pregnancy complications and their management, nutritional assessment of pregnant women, and prevention of mother-to-child HIV transmission) was uncommon in private facilities. This topic is typically covered during in-service training for government facility staff involved in maternal and neonatal care [40]. In the future, the safe motherhood program in Bangladesh should find a way to include private facility workers in the same training.

In Bangladesh, many mothers suffer from anemia, and it is estimated that about 20% of maternal deaths are due to preeclampsia or eclampsia [27]. To diagnose these conditions, hemoglobin and urine protein testing are essential parts of ANC, but they were missing in most union-level public facilities and community clinics, as well as some district and upazila-level public facilities (**Table 2**). The absence of diagnostic tests such as hemoglobin and urine protein tests can result in a tardy diagnosis or non-diagnosis of pregnancy complications like preeclampsia/eclampsia [34]. Moreover, the Ministry of Health should provide financial and technical support to the facilities in setting up laboratory diagnostic services.

Our analysis identified facility type as a significant factor for ANC service preparedness in both the survey years, where union-level facilities, CCs, and private hospitals are less likely to be prepared to offer the service compared with district and upazila public facilities. The low level of ANC service preparedness is due mainly to the shortage of diagnostic capacity (hemoglobin or urine protein tests) at union-level facilities and CCs [26]. The suboptimal preparedness found in other public facilities might be because of an abstruse policy on assigning funds to these facilities that may cause shortages and discrimination in the allocation of medical supplies [41]. Moreover, the leading deficits of private hospitals are the limited availability of ANC guidelines and trained staff [26]. In this study, we observed that the availability of all the tracer indicators for ANC service preparedness was better in district and upazila facilities (**Table 2**). To address the shortage of tracer indicators of ANC service preparedness, private facilities, CCS, and union -level facilities must be strengthened.

Consistent with the prior studies [15, 42], the present study found significant administrative divisions on ANC service preparedness in 2017. This study shows that facilities in the Khulna and Rangpur regions had a higher chance of having medium and high ANC service readiness than those in the Dhaka region. In 2014, almost all divisions were less likely, but in 2017, all other divisions were more likely to be prepared to provide ANC service than the Dhaka division. The observed regional variations in offering ANC services may be due to disparities in health-seeking behavior, service availability, and quality [14]. Further research may be performed to investigate the underlying reasons behind the geographical variation of ANC service among the HCFs in Bangladesh. Nevertheless, we did not observe any significant differences by facility location following a prior study [15].

A health facility must have infection control equipment and supplies suitable to the services delivered, and a lack of adequate infection control items undermines attempts to prevent healthcare-associated infections [43]. Our findings show that infection prevention score is a significant factor for ANC service preparedness, with facilities with a high infection prevention score having a higher likelihood of being prepared to provide the service than their counterparts. This study recommends that HCFs should have enough infection control items to make sure that a pregnant woman gets good care and that the health care workers, visitors, and the rest of the community don't get sick, too.

Visual aids are easy graphical illustrations of numerical expressions that can help everyone, particularly people with low literacy and numeracy skills [44]. The findings of this study reveal a significant association between visual aid for client education and ANC service preparedness, where facilities without visual aid for client education were less likely to be prepared to provide

ANC service. Generating consciousness among childbearing women about the elimination of health-related difficulties during pregnancy through visual aids in a health facility may improve the ANC's service preparedness and finally lessen the burden of pregnancy-related problems. There is a necessity to focus on increasing the availability of visual aids at Bangladeshi health facilities, and they should also be checked regularly.

The ANC card is an essential source of health information that gives every pregnant woman an individual record of her medical and obstetric history over time. The woman is advised to carry the card with her at all times, regardless of whether she moves, and to present it at any health facility [45]. According to our findings, facilities that did not keep individual client cards or records for ANC clients were less likely to be prepared to provide ANC services than their counterparts. It could be because facilities that do not keep ANC cards or records for clients are less likely to improve the availability of ANC services, which is required to assess the facility's readiness to provide ANC services.

## Strengths and limitations

This study has several strengths. First, to our knowledge, this study is the first of this nature in Bangladesh, giving important insight into the preparedness of health facilities to offer ANC service. This study utilized nationally representative samples of HCFs in Bangladesh, and our findings provide essential information about the factors responsible for ANC service preparedness. Second, because SPA data are obtained using a complicated sampling approach, cluster effects and sample weights were accounted for in this study's estimations. Finally, we did a comparative study utilizing the 2014 and 2017 BHFS to see how changes in ANC service preparedness have changed throughout Bangladesh's health facilities.

Results from this study should be taken into account in the context of some limitations. First, this study cannot establish causality because the data was obtained at a certain point in time. A longitudinal study is required to further comprehend the factors influencing the readiness of ANC services. Second, this study did not capture provider-level data that would give a better idea about the preparedness of care from the provider's perspective [15]. Third, our analysis concentrated on health facility preparedness, which is an essential issue but not an assurance of offering quality ANC services [39]. Finally, although SPAs of other countries give information on other measures of quality based on observations of ANC consultations as well as client exit interviews, the BHFSs only give information on service availability and readiness. Future research is needed to assess the quality of ANC services at health facilities in Bangladesh.

## Conclusion

Despite a slight increase in the percentage of facilities providing ANC services from 2014 to 2017, many Bangladeshi health facilities lack ANC guidelines, staff training, and laboratory diagnostic capacity. The overall preparedness to provide ANC services was also low. There was a disparity in the overall preparedness of ANC service provision by facility type. To improve the readiness of the ANC service, government authorities may look at a number of policy options: the focus on union-level facilities, CCs, and private facilities; administrative divisions; the availability of infection control items; maintaining individual client cards or records for ANC clients; and the availability of visual aid for ANC clients.

## Supporting information

**S1 Fig. ANC service preparedness score by the survey years.**
(PDF)

## Acknowledgments

We thank the Demographic and Health Survey for allowing us to use the data.

## Author Contributions

**Conceptualization:** Shariful Hakim, Muhammad Abdul Baker Chowdhury.

**Data curation:** Muhammad Abdul Baker Chowdhury.

**Formal analysis:** Shariful Hakim, Muhammad Abdul Baker Chowdhury.

**Methodology:** Shariful Hakim, Muhammad Abdul Baker Chowdhury.

**Project administration:** Muhammad Abdul Baker Chowdhury.

**Software:** Shariful Hakim, Muhammad Abdul Baker Chowdhury, Md Jamal Uddin.

**Supervision:** Muhammad Abdul Baker Chowdhury, Md Jamal Uddin.

**Validation:** Muhammad Abdul Baker Chowdhury, Md Jamal Uddin.

**Writing – original draft:** Shariful Hakim, Muhammad Abdul Baker Chowdhury.

**Writing – review & editing:** Muhammad Abdul Baker Chowdhury, Zobayer Ahmed.

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
