## [Decision Letter · Decision Letter 0]

14 Oct 2021

PGPH-D-21-00564

Are Bangladeshi health care facilities prepared to provide antenatal care services?  Evidence from two nationally representative surveys

Dear Dr. Uddin

Thank you for submitting your manuscript to PLOS Global Public Health. After careful consideration, we feel that it has merit but does not fully meet PLOS Global Public Health’s publication criteria as it currently stands. Therefore, we invite you to submit a revised version of the manuscript that addresses the points raised during the review process.

Please ensure you respond to all the major and minor points raised by the reviewers.  As you will see, they had differing opinions about the first 3 questions posed to them.  In my opinion, you are in compliance with the all three of them as long as you respond adequately to the detailed comments of  the reviewers.

You will see that both reviewers had substantial concerns about your assessment of ANC readiness both in the way you conceptualized it in the study design and the way you analyzed the results of your research. Please provide detailed responses to these concerns

We look forward to receiving your revised manuscript.

Kind regards,

Joseph J Rhatigan, MD

Academic Editor

Journal Requirements:

1. Please provide separate figure files in .tif or .eps format only, and remove any figures embedded in your manuscript file.  If you are using LaTeX, you do not need to remove embedded figures.

2. If you have no competing interests to declare, please state "The authors have declared that no competing interests exist". 

3. Please note that your Data Availability Statement is currently missing:

1. The repository name

2. The DOI / accession number of each dataset, OR a direct link to access each dataset

If your manuscript is accepted for publication, you will be asked to provide these details on a very short timeline. We therefore suggest that you provide this information now, though we will not hold up the peer review process if you are unable.

Additional Editor Comments (if provided):

Reviewers' comments:

Reviewer's Responses to Questions

**Comments to the Author**

1. Does this manuscript meet PLOS Global Public Health’s publication criteria? Is the manuscript technically sound, and do the data support the conclusions? The manuscript must describe methodologically and ethically rigorous research with conclusions that are appropriately drawn based on the data presented.

Reviewer #1: Partly

Reviewer #2: Yes

2. Has the statistical analysis been performed appropriately and rigorously?

Reviewer #1: Yes

Reviewer #2: I don't know

3. Have the authors made all data underlying the findings in their manuscript fully available (please refer to the Data Availability Statement at the start of the manuscript PDF file)?

Reviewer #1: No

Reviewer #2: Yes

4. Is the manuscript presented in an intelligible fashion and written in standard English?

Reviewer #1: Yes

Reviewer #2: No

5. Review Comments to the Author

Reviewer #1: Thank you for the opportunity to review the manuscript entitled, “Are Bangladeshi health care facilities prepared to provide antenatal care services?  Evidence from two nationally representative surveys.” I appreciate the authors’ use of DHS data in 2014 and 2017 to understand the ability of health facilities to provide antenatal care to pregnant women. These sorts of analyses are important for planning maternal and women’s health services; however, I think the analysis requires some additional clarity and inputs.

MAJOR

1. Please note tables 1 and 2 were cut off in the submission; I could not, therefore, review them fully.

2. The definition of readiness used in the study requires additional clarification. Of the 4 tracer indicators, and 6 individual measures that make up the “readiness score” it is used as a score and sometimes as a percentage throughout the paper. It is confusing and should be streamlined. For example, the authors report in the Methods section that the a score of 100 means the facility is completely prepared; however they report in the Results that the overall score was “low in both the survey years (4.4 and 4.3).” To be clear to the readers, please clarify which score is used, the possible range of the scores and how it should be interpreted.

3. A dichotomization of “prepared or unprepared” to provide antenatal care is challenging when there are only 4 tracer indicators use in the scoring and provision of at least 50% of the indicators defines a facility as “ready.” From a clinical and implementation perspective, this is very concerning that a mere half of the 4 necessary services to be provided in antenatal care would be acceptable. In Haiti and Kenya papers the authors refer for the cutoff (reference 29 and 30), a three-level outcome was used (low, medium, and high readiness). In reference 31, a study from Tanzania, Bintabara et al required that 50% of items in each domain were required before a facility could be considered prepared. The authors of this manuscript focused on Bangladesh need to reconsider their definitions and align with the current literature, or provide justification for a binary cutoff at 50%.

4. As noted in comments 2 and 3, I am concerned with the application of a 50% cutoff to determine a facility as ready to provide antenatal care and how that affects the authors recommendations in the discussion. It would be helpful to reflect the impact of such low preparedness on maternal health in the Discussion.

5. Can the authors provide more details about the choice of the “potential factors” that influenced ANC services preparedness? Several of the potential factors may be considered preparedness in and of themselves and thus appear highly correlated, even defining, the outcome; in particular, one could expect that the infection safety precaution guideline and the basic amenities score would be highly correlated or even define preparedness.

6. Please clarify in Table 4 if the multivariable survey logistic regression model included all the potential factors in a single model and thus, the table reports adjusted Odds Ratio and CI? Pleases add a footnote to the table to make this clear.

7. The discussion provides many good points about the correlation between potential factors and ANC preparedness. Currently, the discussion comments on every factor much like shared in the results. It would be helpful if the authors could connect the different factors to provide a summary.

8. In the discussion section, the paragraph focused on regional versus geographical variations is contradictory; the first sentence states there are significant regional variations and the final sentence says there are not any significant differences. Please clarify.

MINOR

1. Clarity is required around authorship. On the PLoS GPH form there are three authors listed, on the title page, there are four. Zobayer Ahmed, MDS, MSS is not listed on the PLoS GPH submission paperwork. He is listed on the preprint, so I assume this was an oversight.

2. I do note that the paper was submitted to two preprint servers but that was not noted by the authors. I believe preprints are acceptable to PLoS journals and their policies; however, I think it should be noted in the paper that this was submitted as preprint. They can be found at: https://assets.researchsquare.com/files/rs-418861/v1/d7d15650-5781-4c60-8735-2bde6482e60c.pdf?c=1631881237 and https://europepmc.org/article/ppr/ppr314736

3. As the readership of PLoS GPH is wide, it would be helpful for the authors to provide more context in the “facility type” definition. It is unclear what level each of the facilities is and how the facilities connect to each other.

4. Please provide the total number of facilities in each column in tables 2 and 3.

Reviewer #2: This study provides an assessment of facility characteristics linked to ANC service preparedness in Bangladesh in 2014 and 2017. Overall, the study uses an excellent data source and employs the appropriate statistical methods. However, the scope of the paper is limited, and I would have appreciated a more detailed analysis on the ANC service preparedness variable and how it links (or is hypothesized to link) to better health outcomes.

Major Comments

- In the abstract, the authors states that a goal was “to assess whether health care facilities in Bangladesh are prepared to provide ANC service”. Currently, the analysis does not answer this question. I would recommend providing a more detailed discussion on the ANC preparedness index by either comparing the distribution of scores in Bangladesh to other countries or WHO reports on this indicator.

- Why did the authors choose to model preparedness as a binary variable instead of continuous or categorical? I see that other studies have chosen to dichotomize at 50%, but it seems that a lot of information is lost by doing this. The authors may want to consider at least showing the distribution of the raw scores and comparing the distribution of scores by factors using a non-parametric test (e.g. a supplemental table that contains the median and interquartile ranges for ANC preparedness by the factors – similar to Table 3).

Minor Comments

- Was the information for the survey self-reported by the facility or observed by an impartial data collector? If self-reported, this should be listed as a limitation.

- Was the same sampling frame used for both surveys? In addition, what were the strata used for stratified random sampling? I realize this information is provided elsewhere, but it would be helpful to define the strata and any additional clustering.

- I want to clarify that the authors excluded facilities that did not offer ANC services. If that is the case, the language in “Selection of study samples” could be a bit clearer to reflect this.

- Please clarify what is meant by “managing authority”.

- The authors use a cutoff of 50% for the basic amenities and infection prevention score – has this been done in prior literature? Similar to the Major Comment on the ANC preparedness, can the authors provide the distribution of this variable for their study population?

- Do the sampling weights account for non-response?

- Was there any missing data in the outcome or factor variables? If so, how was this handled?

- Does “availability of tracer indicators” mean the “% of health facilities that have information available on the indicator” OR is it the “% of health facilities that provide that service”. The text makes it seem like the former, but I am assuming the authors meant the latter. Please clarify in the text.

- In the Univariate Analysis and Bivariate Analysis results section, the authors report results comparing 2014 and 2017, but from the Introduction and Methods sections, it seems like understanding preparedness by the “factors” is more important – and this is actually what the statistical tests are testing. I would recommend discussing the differences in preparedness within each factor, rather than focusing on changes over time.

- Please include your Stata code for the analysis to ensure this research is reproducible.

6. PLOS authors have the option to publish the peer review history of their article (what does this mean?). If published, this will include your full peer review and any attached files.

**Do you want your identity to be public for this peer review?** For information about this choice, including consent withdrawal, please see our Privacy Policy.

Reviewer #1: No

Reviewer #2: No

---

## [Decision Letter · Decision Letter 1]

6 Apr 2022

PGPH-D-21-00564R1

Are Bangladeshi healthcare facilities prepared to provide antenatal care services?  Evidence from two nationally representative surveys

Dear Dr. Uddin

Thank you for submitting your manuscript to PLOS Global Public Health. After careful consideration, we feel that it has merit but does not fully meet PLOS Global Public Health’s publication criteria as it currently stands. Therefore, we invite you to submit a revised version of the manuscript that addresses the points raised during the review process.

Thank you for your work revising this manuscript. One reviewer felt you had addressed all their comments. The second reviewer had a few new comments based on the revision and suggestions on addressing previous ones.

Please see the attached document for their comments. We look forward to reviewing a revised version of your manuscript

We look forward to receiving your revised manuscript.

Kind regards,

Joseph J Rhatigan, MD

Academic Editor

Journal Requirements:

1. Your co-authors, Shariful Hakim (shakim.sust@gmail.com) and Muhammad Abdul Baker Chowdhury (mchow023@fiu.edu), have not confirmed authorship of the manuscript. We have resent them the authorship confirmation email; however please check that the above email address for them is correct and follow up personally to ensure they confirm. Please note that we cannot pass your manuscript to Production until we have received confirmations from all co-authors. 

2. Please amend your Financial Disclosure statement. If you did not receive any funding for this study, please simply state: “The authors received no specific funding for this work.”

Additional Editor Comments (if provided):

Thank you for your work revising this manuscript. One reviewer felt you had addressed all their comments. The second reviewer had a few new comments based on the revision and suggestions on addressing previous ones.

Please see the attached document for their comments. We hope you can make this minor revision and resubmit.

Reviewers' comments:

Reviewer's Responses to Questions

**Comments to the Author**

1. If the authors have adequately addressed your comments raised in a previous round of review and you feel that this manuscript is now acceptable for publication, you may indicate that here to bypass the “Comments to the Author” section, enter your conflict of interest statement in the “Confidential to Editor” section, and submit your "Accept" recommendation.

Reviewer #1: All comments have been addressed

Reviewer #2: All comments have been addressed

2. Does this manuscript meet PLOS Global Public Health’s publication criteria? Is the manuscript technically sound, and do the data support the conclusions? The manuscript must describe methodologically and ethically rigorous research with conclusions that are appropriately drawn based on the data presented.

Reviewer #1: Yes

Reviewer #2: Yes

3. Has the statistical analysis been performed appropriately and rigorously?

Reviewer #1: Yes

Reviewer #2: Yes

4. Have the authors made all data underlying the findings in their manuscript fully available (please refer to the Data Availability Statement at the start of the manuscript PDF file)?

Reviewer #1: Yes

Reviewer #2: Yes

5. Is the manuscript presented in an intelligible fashion and written in standard English?

Reviewer #1: Yes

Reviewer #2: Yes

6. Review Comments to the Author

Reviewer #1: Thank you for addressing the comments and revising the strategy to be a 3-level indicator of readiness. I also appreciate the updated tables and supplementary table focusing on description of the health system. For Appendix Table 1, you could consider turning this into a histogram would be easier to interpret.

Reviewer #2: Attached

7. PLOS authors have the option to publish the peer review history of their article (what does this mean?). If published, this will include your full peer review and any attached files.

**Do you want your identity to be public for this peer review?** For information about this choice, including consent withdrawal, please see our Privacy Policy.

Reviewer #1: No

Reviewer #2: No

---

## [Editor Report · Decision Letter 2]

17 Jun 2022

Are Bangladeshi healthcare facilities prepared to provide antenatal care services?  Evidence from two nationally representative surveys

PGPH-D-21-00564R2

Dear Dr. Uddin,

We are pleased to inform you that your manuscript 'Are Bangladeshi healthcare facilities prepared to provide antenatal care services?  Evidence from two nationally representative surveys' has been provisionally accepted for publication in PLOS Global Public Health.

Best regards,

Joseph J Rhatigan, MD

Academic Editor